# Ego-centric Predictive Model Conditioned on Hand Trajectories

## Abstract

In egocentric scenarios, anticipating both the next action and its visual outcome is essential for understanding human–object interactions and for enabling robotic planning. However, existing paradigms fall short of jointly modeling these aspects. Vision-Language-Action (VLA) models focus on action prediction but lack explicit modeling of how actions influence the visual scene, while video prediction models generate future frames without conditioning on specific actions—often resulting in implausible or contextually inconsistent outcomes. To bridge this gap, we propose a unified two-stage predictive framework that jointly models action and visual future in egocentric scenarios, conditioned on hand trajectories. In the first stage, we perform consecutive state modeling to process heterogeneous inputs—visual observations, language, and action history—and explicitly predict future hand trajectories. In the second stage, we introduce causal cross-attention to fuse multi-modal cues, leveraging inferred action signals to guide an image-based Latent Diffusion Model (LDM) for frame-by-frame future video generation. Our approach is the first unified model designed to handle both egocentric human activity understanding and robotic manipulation tasks, providing explicit predictions of both upcoming actions and their visual consequences. Extensive experiments on Ego4D, BridgeData, and RLBench demonstrate that our method outperforms state-of-the-art baselines in both action prediction and future video synthesis.

## 1 Introduction

World models (Ha & Schmidhuber, 2018; Wang et al., 2024b; Liu et al., 2024a; Lai et al., 2025) should be capable of simulating how the physical world changes in response to complex environments through integrated processing of language, vision (Chen et al., 2023; Li et al., 2024), and audio (Ghosal et al., 2023; Yang et al., 2023). Humans naturally possess this ability: we can intuitively imagine how a specific physical action (by ourselves or others) will affect objects and the environment around us. In egocentric settings, particularly in human–object interaction tasks (Touvron et al., 2023; Brown et al., 2020; Chowdhery et al., 2022; Zhang et al., 2023), the human hands (or robot arms) are the primary instruments of action. It is therefore crucial to explicitly model the hand's actions as a conditioning factor when predicting future states of the world.

Despite the importance of jointly modeling actions and future vision, most existing approaches have significant limitations. On one hand, vision-language-action (VLA) (Kim et al., 2024) models focus on predicting actions given visual (and textual) inputs without explicit visual prediction. This can hinder an agent's understanding of the consequences of its actions in, for example: dynamic scenes. On the other hand, video prediction and generation methods (Lai et al., 2025; Hu et al., 2023) attempt to foresee future frames but often do so without explicit action conditioning.

To bridge this gap, we propose a unified two-stage **Ego**centric **P**redictive **M**odel (*Ego-PM*) conditioned on hand trajectories, which unifies action prediction with future frame generation. In the first stage, we explicitly predict future hand trajectories via consecutive state modeling of the heterogeneous inputs (visual observations, language, and action history). In the second stage, we introduce causal cross-attention to fuse multi-modal cues, leveraging inferred action signals to guide frame-by-frame future video generation. Importantly, our method does not require any additional annotations at inference time: the model autonomously predicts the necessary hand trajectory as an intermediate

step and then uses it to generate the future video frames. This design makes our approach both powerful and practical for real-world applications.

Our key contributions can be summarized as follows:

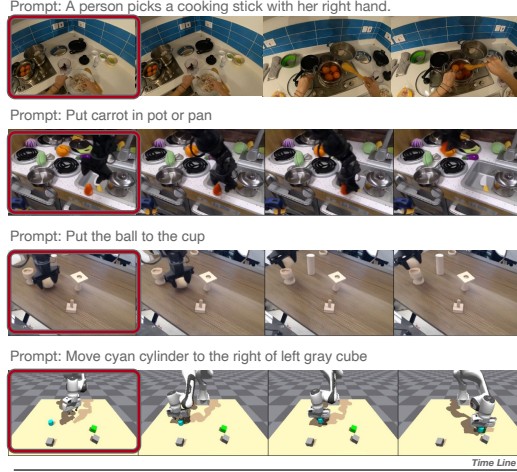

- We introduce the first unified model that explicitly predicts both upcoming actions and future visual frames, addressing tasks that prior approaches handled separately.

- We propose a novel Consecutive State Modeling (CoSMo) strategy to explicitly predict hand trajectories, and introduce a new causal cross-attention scheme to enhance visual synthesis conditions through the intermediate hand action representation. These strategies significantly improve the quality of hand trajectory predictions by leveraging historical information, which in turn leads to more accurate and consistent future frame generation.

- This is the first unified predictive model applicable to both human egocentric and robotic manipulation scenarios, as demonstrated by its strong performance on Ego4D, BridgeData, and RLBench.

Figure 1: Examples of egocentric predictive modeling. Given a sequence of past video frames and text prompt, our model first predicts the upcoming action and then generates future video frames conditioned on that action. Shown are results from (a) Ego4D, (b) and (c) BridgeData, and (d) the RLBench environment.

## 2 RELATED WORK

**Text-Image-Based Video Generation.** Text-image-based video generation has emerged as a promising field that bridges natural language processing (NLP) and computer vision, enabling models to generate video content from textual and visual descriptions. These methods typically process text and images separately using a text tokenizer and vision encoder, after which the tokens are integrated through a unified autoregressive model. A vision decoder then generates images based on the multimodal inputs.

Various approaches employ generative adversarial networks (GANs) or variational autoencoders (VAEs) for text-to-image synthesis (Souček et al., 2024), while others leverage diffusion models (Hu et al., 2023). For instance, GenHowTo (Souček et al., 2024) employs separate models for action state prediction and final state prediction, which generates frames in stages but results in redundant and computationally expensive training. Current text-image-based video generation methods face significant limitations in ego-centric contexts, as they often lack the first-person perspective and fail to incorporate action-conditioned predictions necessary for interactive, immersive experiences.

**Robotics & VLA Models.** LLARVA (Niu et al., 2024) introduces a vision-action instruction tuning approach that uses structured natural language prompts to unify diverse robotic tasks, along with an auxiliary 2D "visual trace" prediction to align vision and action spaces. AdaWorld (Gao et al., 2025) is a world-model pretraining approach that learns to incorporate actions without explicit labels by extracting latent action representations from video. This&That (Wang et al., 2024a) couples deictic language commands ("this/that") with pointing gestures for video generation in robotic planning. TORA (Zhang et al., 2025) adds controllable motion trajectories to a diffusion transformer for text-to-video generation. Notably, our model uses only simple hand position annotations during training as explicit action cues, and it requires no additional inputs at inference—predicting the hand trajectory on its own. Unlike previous trajectory-conditioned video generation approaches that often assume a static background, our model captures dynamic scene context and moving foreground objects, leading to more realistic future video predictions.

**Ego-Centric Vision-Language Models.** Recent efforts in ego-centric vision have focused on understanding human actions and attention (Huang et al., 2020; Sudhakaran et al., 2019; Kazakos et al., 2019; Lai et al., 2023; Luo et al., 2024; Xu et al., 2025; Liu et al., 2024b), modeling hand-object interactions (Ragusa et al., 2023; Liu et al., 2022; Goyal et al., 2022), and estimating human poses (Tome et al., 2020; Wang et al., 2023; Li et al., 2023) from a first-person perspective. In ego-centric video-language modeling, Lin *et al.* (Lin et al., 2022) introduced EgoVLP, a large-scale pretraining model for video-language tasks. For visual generation, Jia *et al.* (Jia et al., 2022) leveraged GANs (Goodfellow et al., 2020) to generate future head motion in a hand forecasting task, while Zhang *et al.* (Zhang et al., 2017) used GANs to anticipate future gaze direction. Ye *et al.* (Ye et al., 2023) developed an affordance diffusion model to generate possible hand-object interactions from an object image in an egocentric view. More recently, LEGO (Lai et al., 2025) synthesized egocentric action frames from a given image and text prompt, but its lack of an action input prevents modeling human–environment interactions, and it struggles to maintain temporal coherence across consecutive frames, especially during dynamic hand movements. These limitations underscore the need for an egocentric world model that integrates multimodal inputs (vision, language, and action) to produce coherent, contextually relevant predictions from a first-person perspective.

# 3 METHODOLOGY

## 3.1 PROBLEM DEFINITION

Given a $n$-timestep egocentric video sequence $\mathcal{V}_{t:0 \to n} = \{v_0, v_1, \ldots, v_n\}$, a text prompt $\mathcal{T}$, and an action sequence $\mathcal{A}_{t:0 \to n}$, Ego-centric Predictive Model (Ego-PM) captures the multimodal context and predicts both future frames and upcoming actions. The goal of Ego-PM can be formulated as:

$$\max p\big(\{\mathcal{V}_{t:>n},\, \mathcal{A}_{t:>n},\, \mathcal{T}_{\mathcal{A}}\} \mid \{\mathcal{V}_{t:0 \to n},\, \mathcal{A}_{t:0 \to n},\, \mathcal{T}\}\big), \qquad (1)$$

where $\mathcal{T}_{\mathcal{A}}$ denotes a textual description of the predicted action. Ego-PM thereby predicts future frames $\mathcal{V}_{t:>n}$, future actions $\mathcal{A}_{t:>n}$, and an action description $\mathcal{T}_{\mathcal{A}}$ conditioned on the observed inputs.

## 3.2 OVERALL ARCHITECTURE

As previously noted, VLA models (Niu et al., 2024; Kim et al., 2024) handle action prediction without visual forecasting, whereas video generation methods (Lai et al., 2025; Hu et al., 2023) ignore action conditioning and may produce future frames inconsistent with the actions taken.

To bridge this gap, we propose a unified **Ego**centric **P**redictive **M**odel (*Ego-PM*) conditioned on hand trajectories, which unifies action prediction with future frame generation. Given the challenges of training convergence and memory limitations, we adopt a two-stage training strategy similar to prior work (Lai et al., 2025; Liu et al., 2024a). In the first stage, we explicitly predict future hand trajectories via consecutive state modeling of the heterogeneous inputs (*i.e.*, visual observations, language, and action history). In the second stage, we introduce causal cross-attention to fuse multimodal cues, leveraging inferred action signals to guide frame-by-frame future video generation. The overall architecture of Ego-PM is illustrated in Fig. 2, and the following sections detail each component as well as our CoSMo strategy for consistent state prediction.

## 3.3 STAGE I: EXPLICIT ACTION MODELING

Stage I explicitly models the upcoming action (*i.e.*, the hand trajectory) using our consecutive state modeling strategy. Specifically, we first align embeddings of the image, text, and action inputs from previous states, and then sequentially predict an enhanced action description as well as an explicit action representation for the next state.

**Vision Encoder.** We use a pre-trained CLIP visual encoder $\Phi_V$ to extract visual features from each frame. To align image features with the text embedding space, we apply a linear projection $\mathcal{F}$, yielding aligned visual tokens $\mathcal{F}(\Phi_V(\mathcal{V}))$ for each frame.

**Action Intent Encoder&Decoder.** To equip the pretrained LLM with action-awareness, we design a lightweight action encoder that integrates action inputs into the token embedding space. We

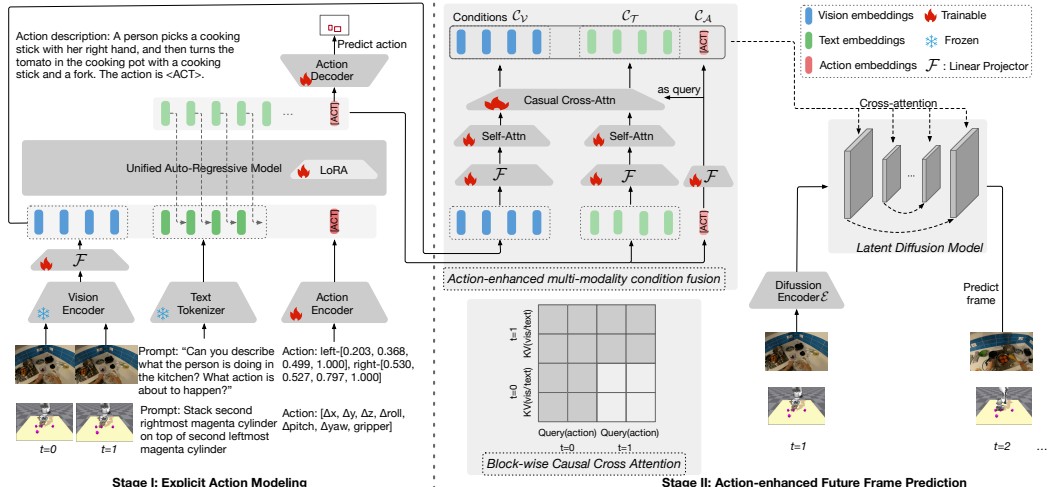

Figure 2: Training framework of Ego-Centric Predictive Model (Ego-PM). In Stage I, a pre-trained LLM is fine-tuned via LoRA (Hu et al., 2022) to process multi-modal inputs (images, text, and actions). Using a Consecutive State Modeling strategy, the autoregressive model explicitly explores the hand trajectory representation by taking two adjacent states as input. In Stage II, a block-wise causal cross-attention mechanism enhances the visual and textual embeddings using the inferred action embedding from Stage I. These combined conditions are then projected into the latent space and used to train a latent diffusion model for predicting future egocentric frames.

introduce a special token $\langle \text{ACT} \rangle$ to indicate the presence of an action, allowing the action embedding $\Phi_A(\mathcal{A})$ to be concatenated with vision and text tokens. During inference, the model's action tokens are decoded via a small MLP-based decoder (matching the encoder) to reconstruct the original action signals. The action intent encoder and decoder are both 3-layer MLPs.

**Autoregressive Model.** We build upon LLaVA (Chen et al., 2023) as the initialized autoregressive model, following a setup similar to LEGO (Lai et al., 2025). At each time step $t$, both text and action tokens are incorporated to enhance temporal understanding. The model is trained on interleaved sequences of vision, text, and action tokens, enabling autoregressive prediction with improved temporal coherence.

**Consecutive State Modeling.** Effective temporal modeling and action–effect interaction are crucial for egocentric predictive modeling. We observe that prior work (Lai et al., 2025) relied solely on the current state to predict the next state, which is suboptimal for sequential predictions. To improve temporal coherence, we introduce a Consecutive State Modeling (CoSMo) strategy. CoSMo leverages both the current and previous states to predict the future state, providing a more robust representation of temporal dynamics. By conditioning on two consecutive time steps (states $t$ and $t-1$) instead of one, the model learns richer temporal dependencies that lead to more coherent and accurate predictions.

### 3.4 STAGE II: ACTION-ENHANCED FRAME PREDICTION

Existing text-based image generation methods (Hu et al., 2023; Souček et al., 2024) and world models (Liu et al., 2024a; Wang et al., 2024b) utilize visual and textual information to predict future frames but lack action guidance. However, we posit that hand trajectory information is essential for human–object interactions. Therefore, explicitly integrating hand trajectory cues into the generation process should yield more detailed and accurate predictions. In this stage, we first enhance the multi-modal conditions (vision and text embeddings) with the inferred hand trajectory embedding. We then fuse these conditions to guide a latent diffusion model, which generates future video frames sequentially.

**Causal Cross Attention.** To predict future frames from historical information, we implement strict causal masking so that *keys/values can be attended by action query tokens from the same*

*or earlier time steps* ($\tau \leq n + 1$). In practice, as illustrated in Fig. 2, we treat the action embedding from Stage I as the Query and perform block-wise CCA over the visual embeddings (as Key and Value). Concretely, in each attention block we use the standard scaled dot-product attention with an additive mask

$$\text{CCA}(Q, K, V) = \text{softmax}\left(\frac{QK^\top}{\sqrt{d}} + M\right)V, \qquad M_{ij} = \begin{cases} 0 & \text{if } j \geq i, \\ -\infty & \text{otherwise.} \end{cases}$$

Here, $i$ indexes query tokens ordered by time (then by modality), and $j$ indexes key/value tokens under the same ordering.

**Action-Enhanced Multi-Modality Condition Fusion.** To form the vision condition for frame generation, we concatenate the current frame ($\mathcal{V}_{t=n}$) and previous frames ($\mathcal{V}_{t:<n}$), apply a linear projection $\mathcal{F}$, and then use self-attention and Causal Cross-Attention (CCA) to capture visual context:

$$\mathcal{C}_\mathcal{V} = \text{CCA}\Big(\text{Self-Attn}\big(\mathcal{F}(\Phi_V(\mathcal{V}_{0\to n}))\big)\Big). \tag{2}$$

For text conditioning, we use CLIP's text encoder $\psi$ to obtain a feature embedding for the prompt $\psi(\mathcal{T})$. We concatenate this with the text embedding generated by the LLM in Stage I (denoted $\mathcal{T}_\mathcal{A}$) and apply self-attention followed by causal cross-attention similar to the vision branch:

$$\mathcal{C}_\mathcal{T} = \text{CCA}\Big(\text{Self-Attn}\big(\mathcal{F}([\psi(\mathcal{T}), \mathcal{T}_\mathcal{A}])\big)\Big). \tag{3}$$

To build the action condition, we project the Stage I action embedding into the diffusion model's latent space using a linear layer $\mathcal{F}$:

$$\mathcal{C}_\mathcal{A} = \mathcal{F}\big(\mathcal{A}_{t:1\to n}\big). \tag{4}$$

Finally, we combine all conditioned inputs: $\mathcal{C} = [\mathcal{C}_\mathcal{V}, \mathcal{C}_\mathcal{T}, \mathcal{C}_\mathcal{A}]$, which is fed into the diffusion model's UNet via cross-attention layers to guide iterative future frame predictions.

**Frame Prediction.** We employ a latent diffusion model (LDM) (Rombach et al., 2022) to generate frames in the latent space of a pre-trained variational autoencoder (with encoder $\mathcal{E}$ and decoder $\mathcal{D}$). We follow the image-based LDM approach, generating future frames one at a time. Starting from the last observed frame $v_n$, we encode it as $z_n = \mathcal{E}(v_n)$ and add noise over diffusion timesteps. The LDM network $\epsilon_\theta$ is trained to predict the added noise $\epsilon$ in $z_t$ at each diffusion step $t$, conditioned on our multimodal context $\mathcal{C}$ (i.e., $\mathcal{C}_\mathcal{V}, \mathcal{C}_\mathcal{T}, \mathcal{C}_\mathcal{A}$).

### 3.5 Training Objectives

**Stage I.** Previous approaches primarily guided predictions with vision and text, overlooking explicit supervision in the action space. To address this, we introduce an objective for consecutive state modeling that includes action supervision, improving the quality of future predictions:

$$\mathcal{L}_{\text{CoSMo}} = \mathcal{L}_{\text{lang,t=1}} + \lambda_1 \mathcal{L}_{\text{act,t=1}} + \mathcal{L}_{\text{lang,t=2}} + \lambda_1 \mathcal{L}_{\text{act,t=2}}, \tag{5}$$

where $\mathcal{L}_{\text{lang}}$ is the standard language modeling loss used in LLaMa (Touvron et al., 2023), and $\mathcal{L}_{\text{act}}$ represents the action loss with a hyperparameter $\lambda_1$.

For human activity dataset (Ego4D (Grauman et al., 2022)), we supervise predicted hand coordinates using a combination of L1 and GIoU (Rezatofighi et al., 2019) losses:

$$\mathcal{L}_{\text{act}}(\hat{b}, b_{\text{gt}}) = \mathcal{L}_{\text{L1}}(\hat{b}, b_{\text{gt}}) + \lambda_2 \mathcal{L}_{\text{GIoU}}(\hat{b}, b_{\text{gt}}), \tag{6}$$

where $\hat{b}$ and $b_{\text{gt}}$ denote predicted and ground-truth hand bounding boxes, respectively, and $\lambda_2$ balances the terms. For robot data (BridgeData (Walke et al., 2023), RLBench (James et al., 2020)), we similarly supervise robot arm keypoints (7D pose) using L1 loss.

**Stage II.** The diffusion process in Stage II is guided by a diffusion-specific loss function, minimizing the following latent diffusion objective:

$$\mathcal{L}_{\text{LDM}} = \mathbb{E}_{\mathcal{E}(\mathcal{V}),\mathcal{C},\epsilon\sim\mathcal{N}(0,1),t}\Big[\|\epsilon - \epsilon_\theta(z_t, t, \mathcal{C})\|_2^2\Big], \tag{7}$$

where $\epsilon$ represents the added noise, and $z_t$ is the noisy latent at timestep $t$. This loss encourages the model to denoise $z_t$ based on multimodal conditioning $\mathcal{C}$, yielding temporally coherent and contextually accurate predictions.

| Baseline | Frame Prediction | | | | | | Action Prediction |
|---|---|---|---|---|---|---|---|
| | EgoVLP | EgoVLP$^+$ | CLIP | FID$\downarrow$ | PSNR | LPIPS$\downarrow$ | Hand IoU |
| LWM$^*$ | 62.31 | 76.54 | 77.89 | 24.95 | 11.73 | 38.12 | - |
| LEGO | 65.65 | 80.44 | 80.61 | 23.83 | 12.29 | 36.43 | - |
| Ours | **69.11** | **83.98** | **83.24** | **21.31** | **16.12** | **33.23** | **44.25** |

Table 1: Ego4D single-step prediction results ($t = 2$ predicted from $t = 1$). We report frame metrics (EgoVLP, EgoVLP$^+$, CLIP, FID, PSNR, LPIPS) and action metric (Hand IoU). Lower values are better for metrics marked with $\downarrow$. *For a fair comparison, we finetune LWM with the same egocentric dataset to minimize the domain gap.

| Baseline | Frame Generation | | | | | | Action Prediction |
|---|---|---|---|---|---|---|---|
| | EgoVLP | EgoVLP$^+$ | CLIP | FID$\downarrow$ | PSNR | LPIPS$\downarrow$ | Hand IoU |
| LWM$^*$ | 61.13 | 75.43 | 76.47 | 25.23 | 11.23 | 38.94 | - |
| LEGO | 62.34 | 77.31 | 77.45 | 25.23 | 11.24 | 37.76 | - |
| Ours | **65.87** | **81.13** | **81.24** | **22.59** | **13.89** | **34.32** | **40.71** |

Table 2: Ego4D consecutive prediction results ($t = 3$ predicted using $t = 1$ and the intermediate prediction for $t = 2$). The same metrics are reported as in Table 1. Bold numbers indicate the best performance, and $\downarrow$ indicates lower is better.

## 4 EXPERIMENTS

We evaluate Ego-PM on both human egocentric (Ego4D) and robotic (BridgeData V2, RLBench) scenarios. We first describe our datasets and metrics, then present quantitative and qualitative results comparing Ego-PM to state-of-the-art baselines. We also provide ablation studies to examine the contributions of CoSMo and the action conditioning.

### 4.1 DATASET

**Human Activity (Ego4D).** We use the Ego4D egocentric video dataset (Grauman et al., 2022) focusing on the Pre-Conditioning (PRE-15) and Point-of-No-Return (PNR) moments. We define the start state $t = 0$ at PRE-15, the intermediate state $t = 1$ at PNR, and the final state $t = 2$ as the frame after PNR (same time interval as between PRE-15 and PNR). The text narration serves as the prompt, and hand position tracks are used as action inputs during training.

**Robot Manipulation (BridgeData & RLBench).** We train and evaluate on (i) BridgeData V2 (Walke et al., 2023), a human-to-robot demonstration dataset, and (ii) additionally test on RL-Bench (James et al., 2020), a simulated robotic manipulation benchmark.

For BridgeData, we use the provided multi-step human demonstration videos and textual instructions. For RLBench, 3D-based robotic manipulation methods (Goyal et al., 2023) reconstruct 3D scenes, render multi-view images, predict heatmaps, and then project these into 3D action coordinates. Although these pipelines achieve high performance, they rely on costly 3D annotations, accurate calibration, and computationally expensive rendering. Our method aims to provide a lighter-weight alternative by leveraging existing 2D annotations (hand locations for ego-centric videos and arm positions for robot data), avoiding the need for additional 3D supervision (like depth or point clouds) or heavy pre-processing. Specifically, our model receives *simulator-rendered RGB observations*—i.e., images sampled from the virtual environment's camera(s)—as its visual inputs at each step. Task success is computed with RLBench's standard success detectors for the nine multi-task settings considered(e.g., "open drawer", "push buttons").

### 4.2 METRICS

For future frame prediction, we report both perceptual and pixel-level metrics: EgoVLP and EgoVLP$^+$ alignment scores (Lin et al., 2022), CLIP image-text similarity, Fréchet Inception Distance (FID) (Heusel et al., 2017), Structural Similarity Index (SSIM), Peak Signal-to-Noise Ratio

| Methods | Type | Frame Prediction | | | | | Action Prediction |
| | | FID↓ | FVD↓ | PSNR↑ | SSIM↑ | LPIPS↓ | Success Rate↑ |
|---|---|---|---|---|---|---|---|
| SVD | Generation | 29.49 | 657.49 | 12.47 | 0.334 | 39.1 | - |
| StreamingT2V | Generation | 42.57 | 780.81 | 11.35 | 0.324 | 50.4 | - |
| DragAnything | Generation | 34.38 | 764.58 | 12.76 | 0.364 | 46.6 | - |
| This&That | Generation | 17.28 | 84.58 | **21.71** | 0.787 | **11.2** | - |
| OCTO | VLA | - | - | - | - | - | 20.0 |
| OpenVLA | VLA | - | - | - | - | - | 70.6 |
| Ours | Omni | **16.36** | **83.43** | 21.56 | **0.791** | 11.3 | **73.7** |

Table 3: Quantitative comparison of different methods on the BridgeData V2 dataset. Frame prediction metrics include FID, FVD, PSNR, SSIM, and LPIPS, where lower FID/FVD/LPIPS and higher PSNR/SSIM indicate better performance. Generation-based models do not predict action trajectories (no action results in top block). VLA-based models do not generate frames (no visual metrics). Our model supports both future frame generation and action prediction.

| Method | Extra label | open drawer | meat off grill | turn tap | put money | push buttons | sweep dustpan | slide block | close jar | screw bulb | Avg. Rate |
|---|---|---|---|---|---|---|---|---|---|---|---|
| Image-BC (ViT) | 2D trajectories | 0 | 0 | 16 | 0 | 0 | 0 | 0 | 0 | 16 | 3.56 |
| LLARVA | 2D trajectories | 60 | 80 | 56 | 44 | 56 | 84 | 100 | 28 | 8 | 57.33 |
| RVT | 3D point cloud | 71.2 | 88 | 93.6 | 40 | 100 | 72 | 81.6 | 52 | 32 | 70.04 |
| SAM-E | 3D point cloud | 82.4 | 95 | 100 | 45 | 100 | 100 | 95 | 82 | 30 | **81.04** |
| Ours | No need | 64 | 70 | 60 | 56 | 56 | 84 | 96 | 32 | 12 | **58.89** |

Table 4: Success rates (%) on the RLBench Multi-Task benchmark. Each method is evaluated on 25 episodes per task, with each episode scored as 100 (success) or 0 (failure). We report the average success rate per task for each method.

(PSNR), and Learned Perceptual Image Patch Similarity (LPIPS) (Zhang et al., 2018). Lower FID and LPIPS and higher SSIM and PSNR indicate better visual fidelity.

For action prediction, we use hand mask Intersection-over-Union (IoU) for Ego4D ( predicted vs. ground-truth hand regions) and Success Rate for BridgeData/RLBench (the percentage of successful task completions). Higher values indicate better action prediction accuracy.

## 4.3 CONSECUTIVE STATE PREDICTION

We choose LWM (Liu et al., 2024a) and LEGO (Lai et al., 2025) as benchmarks for both quantitative and qualitative comparisons. Table 1 and Table 2 show Ego4D results for single-step and consecutive predictions, respectively. Our Ego-PM significantly outperforms LWM and LEGO across all metrics. Notably, LEGO (the strongest baseline) lacks the ability to predict actions (Hand IoU not applicable), whereas Ego-PM predicts hand trajectories with high accuracy (44.25 IoU) while also achieving superior frame generation quality (e.g., FID 21.31 vs 23.83 for LEGO in single-step).

For the harder consecutive prediction (Table 2), all methods see performance drops, but Ego-PM maintains a clear advantage in every metric, highlighting its robustness in compounding predictions.

## 4.4 ROBOTIC MANIPULATION RESULTS

Our model is not only effective on human-centric datasets such as Ego4D, but also generalizes well to robot manipulation datasets, including BridgeData V2 (Walke et al., 2023) and RLBench (James et al., 2020). To evaluate its versatility, we compare our model with two representative VLA-based models (OpenVLA (Kim et al., 2024) and OCTO (Team et al., 2024)) and four state-of-the-art video generation models (SVD (Blattmann et al., 2023), StreamingT2V (Henschel et al., 2025), DragAnything (Wu et al., 2024), and This&That (Wang et al., 2024a)).

| Row | State | Component | | | Frame Prediction | | | | | | Action Prediction |
|---|---|---|---|---|---|---|---|---|---|---|---|
| | | CoSMo | Action input | With CCA | EgoVLP | EgoVLP$^+$ | CLIP | FID↓ | PSNR | LPIPS↓ | Hand IoU |
| 1 | $t=2$ | ✗ | Plain text | ✗ | 65.76 | 81.03 | 80.01 | 23.71 | 12.36 | 36.89 | 39.71 |
| 2 | $t=2$ | ✗ | Enc&Dec | ✗ | 67.12 | 82.31 | 81.54 | 22.88 | 13.01 | 35.89 | 40.54 |
| 3 | $t=2$ | ✓ | Enc&Dec | ✗ | 68.32 | 83.44 | 82.53 | 22.12 | 14.10 | 34.31 | 41.43 |
| 4 | $t=2$ | ✓ | Enc&Dec | ✓ | 69.11 | 83.98 | 83.24 | 21.31 | 16.12 | 33.23 | 44.25 |
| 5 | $t=3$ | ✗ | Plain text | ✗ | 62.71 | 77.56 | 77.58 | 25.45 | 11.35 | 37.65 | 36.21 |
| 6 | $t=3$ | ✗ | Enc&Dec | ✗ | 63.15 | 78.53 | 79.13 | 24.34 | 12.56 | 36.56 | 37.36 |
| 7 | $t=3$ | ✓ | Enc&Dec | ✗ | 64.78 | 80.21 | 80.31 | 23.89 | 12.87 | 35.21 | 38.78 |
| 8 | $t=3$ | ✓ | Enc&Dec | ✓ | 65.87 | 81.13 | 81.24 | 22.59 | 13.89 | 34.32 | 40.71 |

Table 5: Ablation study on Ego4D evaluating key components of Ego-PM: Consecutive State Modeling (CoSMo), action input format, and Causal Cross-Attention (CCA). Frame metrics include EgoVLP, EgoVLP$^+$, CLIP, FID, PSNR, and LPIPS (higher is better for EgoVLP/EgoVLP$^+$/CLIP/PSNR, lower is better for FID/LPIPS). Hand IoU measures action prediction (higher is better). Configs 1–4 use one-step prediction ($t = 2$), and 5–8 use consecutive two-step prediction ($t = 3$).

On BridgeData (Table 3), our model is the only "omni-capable" approach that excels in both frame generation and action prediction. VLA baselines (OpenVLA, OCTO) predict actions well but cannot generate frames; generation models (SVD, StreamingT2V, etc.) produce frames but lack action outputs. Ego-PM achieves the highest action success rate (73.7% vs 70.6% OpenVLA), the best FID/FVD/SSIM, and comparable PSNR/LPIPS.

Table 4 reports RLBench multi-task success rates. Although some methods like SAM-E and RVT have higher overall success by leveraging heavy 3D inputs, our model (which uses no additional 3D sensors or ground-truth trajectories) still outperforms 2D-trajectory baselines and is competitive with methods relying on richer inputs. For tasks such as "sweep dustpan" and "slide block", Ego-PM matches or exceeds the performance of more complex baselines, despite its simpler input requirements. This indicates a favorable trade-off: Ego-PM achieves strong generalization with far less annotation overhead.

## 4.5 ABLATION STUDIES

To analyze the contributions of our design choices, we perform controlled experiments on Ego4D. Table 5 summarizes the results.

**Effect of Action Encoding.** In table 5, comparing row 1 vs. 2 (and 5 vs. 6), we see that replacing the naive "action as plain text" input with our dedicated action encoder–decoder yields notable improvements (e.g., Hand IoU from 39.71 to 40.54, FID from 23.71 to 22.88 at $t = 2$). This validates that treating the action modality separately (instead of as additional text) is important. The plain-text approach fails to convey distinct action information to the model, whereas our encoded action tokens provide effective guidance.

**Effect of CoSMo** Row 3 (and 7) introduce our Consecutive State Modeling strategy. Using two consecutive states during prediction (row 3) improves all metrics over row 2 (single-state), confirming that CoSMo enhances temporal coherence. EgoVLP$^+$ (video-text alignment) and PSNR both rise with CoSMo, and Hand IoU improves as well (from 40.5 to 41.4). Intuitively, looking at both $t = 1$ and $t = 0$ provides the model with motion context, leading to better anticipation of $t = 2$.

**Effect of Causal Cross-Attention.** Row 4 (and 8) add the causal cross-attention (CCA) fusion. This yields the best performance: e.g., Hand IoU jumps from 41.4 to 44.3 at $t = 2$, and FID drops from 22.12 to 21.31. CCA particularly helps maintain quality into the second predicted frame ($t = 3$), as seen by improvements from Row 7 to 8 (e.g., EgoVLP$^+$ 80.2 to 81.1, Hand IoU 38.8 to 40.7). This indicates that aligning the visual and textual tokens with the action query in a causal manner (ensuring proper temporal order) benefits both action and frame prediction.

**Consecutive Predictions** To test the model's ability to handle consecutive predictions, we use the predicted state at $t = 2$ to predict the next state at $t = 3$. The results, presented in rows 4-6 of Table 5, show a noticeable performance drop across all baselines. This indicates that existing

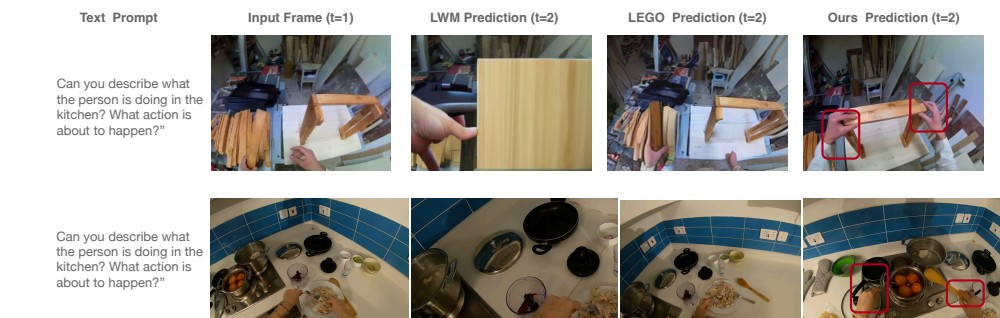

Figure 3: Qualitative comparisons of generated future frames (with predicted actions) for a sample egocentric scenario.

world models struggle with consecutive predictions due to compounding errors in earlier state predictions. However, our model consistently outperforms the baselines, showcasing its robustness and effectiveness in handling consecutive predictions.

### 4.6 VISUALIZATION

We present qualitative results in Fig. 3, comparing Ego-PM to baseline models for an Ego4D example. Even after egocentric fine-tuning, LWM retains biases from its exocentric pretraining and design, which limit its ability to capture fine-grained egocentric hand–object interactions. LEGO (Lai et al., 2025) does better initially but fails to render the hand properly and drifts by the second predicted frame. Ego-PM, however, generates a realistic hand interacting with the correct object (per the text prompt) and preserves the scene context (background remains consistent). Moreover, Ego-PM is the only model that provides an explicit hand trajectory prediction (green overlay), indicating where the hand will move. These visualizations underscore how Ego-PM more holistically understands "what will happen next" in first-person scenes.

## 5 CONCLUSION

In summary, we have presented a novel egocentric predictive model conditioned on hand trajectories, capable of both action prediction and future video generation. By explicitly incorporating hand motion as an intermediate representation, our approach addresses the limitations of prior methods that separated action reasoning from visual forecasting. The proposed model not only learns to anticipate which action will occur in a first-person scenario, but also visualizes the consequences of that action in the form of future frames – all within a unified architecture.

Our contributions include introducing a simple yet powerful conditioning signal (hand position) to guide the learning of action–effect relationships, and a consecutive state modeling strategy that effectively integrates heterogeneous inputs. Crucially, our model operates without requiring any manual action annotations at test time, making it practical for real-world deployment. Extensive experiments on three distinct datasets – Ego4D (human egocentric videos), BridgeData (human-to-robot scenarios), and RLBench (robotic manipulation tasks) – validate the effectiveness of our model. It consistently outperforms state-of-the-art baselines in both the accuracy of predicted actions and the quality of generated future video sequences. These results underscore the generality of our approach, as the same model can be seamlessly applied to both human and robotic perspectives.

Future work can build upon this foundation by exploring even richer conditioning signals and by extending the model to handle longer-horizon predictions and more complex multi-agent interactions. Overall, our hand-conditioned egocentric predictive model opens new avenues for intuitive and interactive AI systems that understand and predict the coupling between actions and their effects in the world.

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

## A  LLM Use Disclosure

We used ChatGPT solely for **language polishing (grammar and style)** in the *Introduction*, *Related Work*, and selected table captions. The model did not generate technical content, methods, results, or claims. All edits were reviewed by the authors, who accept full responsibility for the content of this submission.

## B  Ethics Statement

Our work uses publicly released datasets for egocentric video and robot manipulation (e.g., Ego4D, RLBench, BridgeData V2) under their respective licenses. We rely on the dataset providers' consent and curation protocols and do not attempt to identify individuals or extract sensitive attributes. We release no additional personally identifying annotations.

## C  Reproducibility

We provide: preprocessing scripts and file lists for each dataset/split; model configs with architecture hyperparameters (context length, tokenizers, LoRA ranks), optimizer settings, learning-rate schedules, batch sizes, image resolution, and augmentations; training/inference commands; and evaluation scripts that compute PSNR, SSIM, LPIPS (normalized to $[0, 1]$), FVD (I3D-based), CLIP similarity, and task Success Rate with unweighted task means and $95\%$ CIs over three seeds. **Code and checkpoints will be released under a permissive license upon publication**.

## D  Appendix I: Related Work

In the main paper, we briefly summarize relevant research on text-based video generation, robotics and vision-language-action (VLA) models, and egocentric vision-language models. In addition to these areas, we also provide a discussion of related *world model* approaches.

**World Models.**  World models (Ha & Schmidhuber, 2018) enable intelligent agents to understand, predict, and make decisions based on interactions with their environments. In computer vision, world models are widely used for video prediction (Lotter et al., 2017; Lee et al., 2018), which is critical for downstream tasks such as autonomous driving (Hu et al., 2023; Shi et al., 2023; Zheng et al., 2025; Tian et al., 2024) and robotics (Liang et al., 2022; Lu et al., 2025; Firoozi et al., 2023). These models generate plausible future frames by conditioning on the agent's current state, actions, and environmental context, thereby enhancing temporal coherence and action prediction accuracy.

LWM (Liu et al., 2024a) tackles long-sequence prediction via masked sequence packing to improve training efficiency. However, it directly concatenates video frames without explicitly modeling consecutive states. GAIA-1 (Hu et al., 2023) predicts future frames conditioned on visual inputs and action sequences, facilitating real-time event anticipation in autonomous driving. Despite their effectiveness, GAIA-1 and related models (Wang et al., 2024b) primarily operate from exocentric (third-person) viewpoints, which limits their capacity to model fine-grained egocentric details such as hand movements. While world models have made considerable progress, they still face challenges in egocentric and interactive tasks where a user-centered, first-person view is essential for immersive applications.

## E  Appendix II: Technical Details

**Overview of Ego-PM.**  An overview of the Ego-Centric Predictive Model (Ego-PM) is illustrated in Fig. 4, inspired by the architecture of the original World Model (Ha & Schmidhuber, 2018). The model consists of three core components:

- *Multimodal Encoders* extract features from visual, textual, and action inputs. Incorporating the action space is critical for accurate prediction.

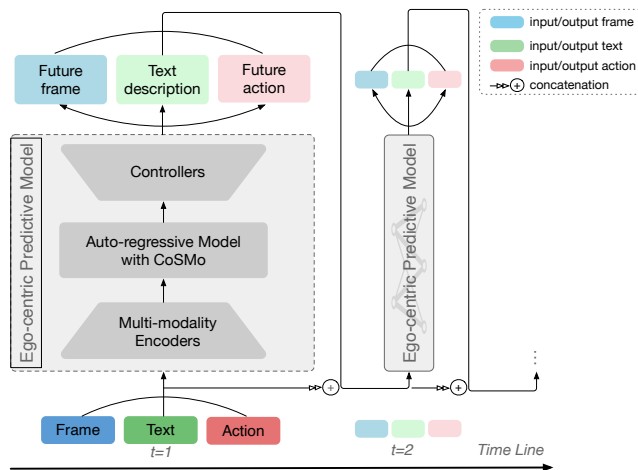

Figure 4: Overview of Ego-PM architecture, comprising multimodal encoders, an autoregressive model, and controllers. The model predicts consecutive future states, including both frames and actions. Darker colors denote input states, and lighter colors indicate predicted states.

| Model | Input Modality | | | Prediction Capability | | |
|---|---|---|---|---|---|---|
| | Scene | Text | Action | Frame | Text | Action |
| LWM | Exocentric | ✓ | | ✓ | ✓ | |
| GAIA-1 | Exocentric | ✓ | Vehicle behavior | ✓ | | |
| WorldDreamer | Exocentric | ✓ | Vehicle behavior | ✓ | | |
| LEGO | Egocentric | ✓ | | ✓ | ✓ | |
| OpenVLA | Robot | ✓ | | | ✓ | |
| PEVA | Human | ✓ | Body skeleton | ✓ | | |
| **Ours (Ego-PM)** | Omni | ✓ | Hand pose / Robot kinematics | ✓ | ✓ | ✓ |

Table 6: Comparison of baselines with respect to input modality and prediction capabilities. Our model uniquely supports egocentric inputs with multimodal conditions (text, visual, action) and predicts future frames, narrative text, and action movements.

- *An Autoregressive Model* learns cross-modal correlations to enhance temporal coherence and explicitly predict future hand trajectories.

- *A Controller* uses these representations to predict future frames, generate narrative descriptions, and anticipate upcoming actions.

**Training Details.** We use Bi-State Prediction to illustrate the CoSMo (Consecutive State Modeling) strategy. Given triplets of input data $\{\mathcal{V}_{t:1\rightarrow 2}, \mathcal{A}_{t:1\rightarrow 2}, \mathcal{T}\}$, the model learns to predict future states from consecutive inputs. We introduce a [MASK] token to represent a zero-initialized state at $t = 0$. The model first predicts the state at $t = 2$ using inputs from $t = 0$ and $t = 1$, then continues with $t = 1$ and $t = 2$ to predict $t = 3$, and so on.

**Implementation Details.** By default, we use a 2-timestep history as input (two past frames and their actions). The auto-regressive model is initialized from LEGO's pre-trained weights (or Open-VLA) for Ego4D (or BridgeData). We freeze the CLIP image encoder and fine-tune the projection layer and the LLM for 3 epochs using cross-entropy loss. The LDM is initialized from a Stable Diffusion checkpoint (Rombach et al., 2022). All frames are $256 \times 256$ resolution. We set $\lambda_1 = 0.1$ and $\lambda_2 = 0.01$ in our losses based on validation performance.

About the Hand annotation in Ego4D dataset, we select the subset of Ego4D that contains bounding-box-level hand annotations. About 95% of the selected frame tuples contain at least one visible hand. For frames with no visible hands, we pass a null value to the action encoder.

| Dataset | Dim | Units | Normalization | Supervision |
|---|---|---|---|---|
| Ego4D (hands) | 4 | pixels $(c_x, c_y, w, h)$ | $[0, 1]$ by $(W, H)$ | L1 + GIoU (boxes) |
| RLBench | 7 | $\Delta$pose $(\Delta x, \Delta y, \Delta z, \Delta r, \Delta p, \Delta y)$ + gripper | z-score per-axis over train set | L1 (pose) + BCE (gripper) |
| BridgeData V2 | 7 | $\Delta$pose + gripper | z-score per-axis over train set | L1 + BCE |

Table 7: **Action specifications and supervision.** We standardize units and normalization across datasets. Box IoU is the default for Ego4D; mask IoU is reported when masks are available.

| Model | Frame Prediction | | |
|---|---|---|---|
| | FID↓ | PSNR↑ | LPIPS↓ |
| PVEA-B (original, DiT-based) | 74.338 | 16.01 | 33.7 |
| PVEA-B (reproduced) | 78.56 | 15.61 | 35.8 |
| **Ours (LDM-based)** | **76.21** | **16.10** | **33.4** |

Table 8: Frame prediction results on the Nymeria dataset. Lower FID and LPIPS, and higher PSNR indicate better performance.

**Comparison of Model Capabilities.**    Table 6 compares various models in terms of input modality and prediction ability. LWM (Liu et al., 2024a) can process long exocentric videos and generate both frames and text. GAIA-1 (Hu et al., 2023) and WorldDreamer (Wang et al., 2024b) use exocentric views and vehicle behavior for frame prediction. LEGO (Lai et al., 2025), though not a full world model due to missing action prediction, remains a strong baseline for egocentric frame generation.

**Action specifications across datasets.**    The notion of an "action" varies depending on the environment: egocentric video datasets provide hand-localization signals, while robot manipulation datasets provide low-level control trajectories. To ensure clarity and reproducibility, Table 7 summarizes the action representations we adopt for each benchmark, including their dimensionality, physical units, normalization scheme, and supervision signals. This unified specification allows our model to handle both 2D hand boxes in human videos and continuous pose–gripper commands in robotic environments within the same tokenization framework.

### E.1    Pseudo-code for Consecutive State Modeling (CoSMo)

To improve temporal consistency and action prediction in egocentric settings, we introduce CoSMo, which utilizes both current and previous states to predict future ones. Unlike prior work (Lai et al., 2025) that relies on only the current state, our method captures richer temporal dependencies.

Prompt: Take the blue rectangular box and put in the top left of the table.

Prompt: Close the drawer.

Prompt: Stack right green cube on top of left green cube.

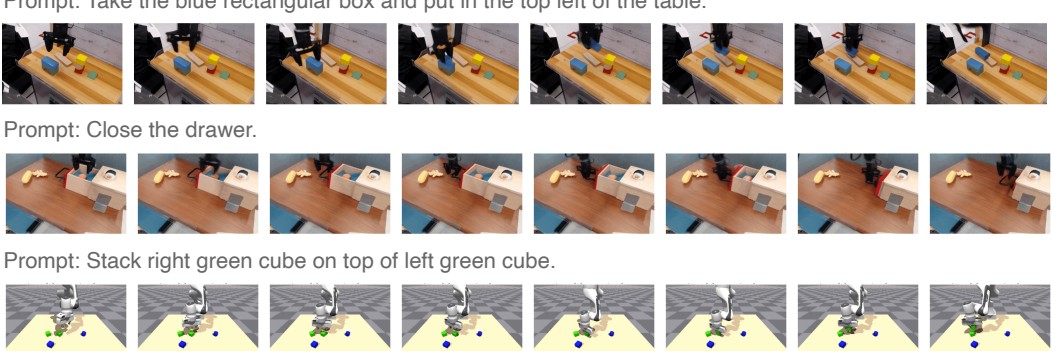

Figure 5: Qualitative comparisons of generated future frames and predicted actions on robotic manipulation benchmarks.

## F    Appendix III: Additional Experiments

**Long-Horizon Prediction.** For longer-horizon prediction, we provide qualitative rollouts Figure 5 on BridgeData and RLBench using Ego-PM. These clips illustrate that our model can generate longer sequences with coherent hand–object interactions.

**Discussion about mutual benefit of joint action and frame prediction.** (1) Action benefits frames: In Table 5, comparing "Plain text" vs. "Enc&Dec" (rows 1 vs. 2 and 5 vs. 6), replacing naive text-only action tokens with an explicit structured action encoder–decoder consistently improves frame metrics (e.g., FID, PSNR, LPIPS) as well as Hand IoU. This indicates that better action modeling directly leads to higher-quality future frames. (2) Frames benefit action: Our CoSMo loss jointly supervises language and action at t=1 and t=2. When we enable CoSMo (rows 2 to 3 and 6 to 7 in Table 5), both frame metrics and Hand IoU improve, showing that the joint temporal modeling of future frames and action trajectories helps the action head become more accurate and temporally consistent.

**Human-Skeleton-Based Video Prediction.** Beyond hand trajectory modeling, we explore the use of full-body motion as an action representation. Specifically, we replace the original $4 \times 2$ hand coordinates with 17 human keypoints (skeleton) to examine whether explicitly modeling richer action space improves video generation.

We use the Nymeria dataset (Ma et al., 2024), which contains egocentric RGB frames, 3D body pose annotations, and textual descriptions in real-world scenarios. We use PEVA (Bai et al., 2025) as the baseline and filter training samples with available text.

PEVA is a video-based DiT method that requires 3–15 continuous frames and corresponding skeleton inputs to predict 64 future frames. In contrast, our method (LDM-based) only requires two frames and a text prompt. We first predict the action representation, then generate future video frames iteratively using action embeddings.

Results in Table 8 show that our method outperforms the PEVA baseline using the same training set, demonstrating that explicit action modeling enhances prediction quality.

---

**Algorithm 1** Consecutive State Modeling (CoSMo)

---

1: **Input:** Video frames $\mathcal{V}_{t:1\rightarrow2}$, Actions $\mathcal{A}_{t:1\rightarrow2}$, Text prompt $\mathcal{T}$
2: **Output:** Predicted frames $\hat{\mathcal{V}}_{t:>2}$, Actions $\hat{\mathcal{A}}_{t:>2}$, Action description $\mathcal{T}_{\mathcal{A}}$
3: Initialize masked inputs: $v_0 \leftarrow [\text{MASK}], a_0 \leftarrow [\text{MASK}]$
4: **for** each training step in Stage I **do**
5:     Sample $\{\mathcal{V}_{t:1\rightarrow2}, \mathcal{A}_{t:1\rightarrow2}, \mathcal{T}\}$
6:     Predict $\{\hat{v}_2, \hat{a}_2, \mathcal{T}_{\mathcal{A}}\}$ using $(v_0, v_1, a_0, a_1, \mathcal{T})$
7:     Predict $\{\hat{v}_3, \hat{a}_3\}$ using $(v_1, \hat{v}_2, a_1, \hat{a}_2, \mathcal{T})$
8:     Compute loss $\mathcal{L}_{\text{CoSMo}}$ and update model parameters
9: **end for**
10: **for** each training step in Stage II **do**
11:     Sample $\{\mathcal{V}_{t:1\rightarrow2}, \mathcal{A}_{t:1\rightarrow2}, \mathcal{T}, \mathcal{T}_{\mathcal{A}}\}$
12:     Predict $\hat{v}_2$ using $(v_0, v_1, a_0, a_1, \mathcal{T}, \mathcal{T}_{\mathcal{A}})$
13:     Predict $\hat{v}_3$ using $(v_1, \hat{v}_2, a_1, \hat{a}_2, \mathcal{T}, \mathcal{T}_{\mathcal{A}})$
14:     Compute loss $\mathcal{L}_{\text{LDM}}$ and update model parameters
15: **end for**

---

