# OpenReview forum: "Ego-centric Predictive Model Conditioned on Hand Trajectories"
_ICLR.cc/2026/Conference — ICLR 2026 Conference Withdrawn Submission_

### Official Review · Reviewer_LcVE · 2025-10-29

**Soundness:** 3
**Presentation:** 3
**Contribution:** 3
**Rating:** 4
**Confidence:** 4

**Summary:**

The paper proposes Ego-PM, a unified two-stage egocentric predictive model designed to jointly predict actions and generate future visual frames, conditioned on hand trajectories. Extensive experiments have demonstrated the effect of each proposed component on multiple real-world and robotic datasets.

**Strengths:**

1. This paper combines action modeling with video generation in a single framework.

2. This model demonstrates consistent improvements across multiple benchmarks (Ego4D, BridgeData, RLBench) and modalities.

**Weaknesses:**

1. One of the main contribution of this paper is the joint prediction of action and future frames, however, I do not see the qualitative/quantitative analysis showing the mutual benefit.

2. It is not clear how the model fuses multi-modal information when not using CCA in Table 5.

3. The text format could be further improved (L84-L85).

4. This paper misses some discussions with recent works on conditioned egocentric image/video generation [1][2][3].

[1] Luo et al. "Put myself in your shoes: Lifting the egocentric perspective from exocentric videos." ECCV 2024.
[2] Xu et al. "Egoexo-gen: Ego-centric video prediction by watching exo-centric videos." ICLR 2025.
[3] Liu et al. "Exocentric-to-egocentric video generation." NeurIPS 2024.

**Questions:**

1. In the experiments, the authors explored using one or two consecutive frames as context, what about using more time steps?

2. What is the advantage of CCA compared to bi-directional attention in your model?

3. The ablation study on consecutive predictions is not very clear. Please explain in detail.

4. In L456-457, the authors claimed the failure of LWM as its exocentric bias, however, the authors already fine-tuned LWM on egocentric data (L260). It is a bit confusing.

---

> ### Author Response · Authors · 2025-11-20
>
> **Q1:  Mutual benefit of joint action + frame prediction**
>
> **A:** Thank you for pointing this out. We agree that the mutual benefit of joint prediction should be made clearer, and we will explicitly highlight it in the revision:
>
>
>
> - **From action → frames.** In Table 5, comparing “Plain text” vs. “Enc&Dec” (rows 1 vs. 2 and 5 vs. 6), replacing naive text-only action tokens with an explicit structured action encoder–decoder consistently improves *frame* metrics (e.g., FID, PSNR, LPIPS) as well as Hand IoU. This indicates that better action modeling directly leads to higher-quality future frames.
> - **From joint objective → better action.** Our CoSMo loss (Eq. (5)) jointly supervises language and action at t=1 and t=2. When we enable CoSMo (rows 2 → 3 and 6 → 7 in Table 5), both **frame metrics and Hand IoU** improve, showing that the joint temporal modeling of future frames and action trajectories helps the action head become more accurate and temporally consistent.
>
>
>
>
>
> In the revised version, we will add an explicit discussion of this mutual benefit.
>
>
>
> ------
>
>
>
> **Q2: How multi-modal fusion works without CCA (Table 5)**
>
>
>
> **A:** Thank you for noting this. In the configurations **without CCA** in Table 5, we first apply a **linear projection + self-attention** for each modality (vision and text). The resulting modality-specific features are then **concatenated** with the projected action embedding and passed as conditioning tokens to the diffusion UNet via its standard cross-attention layers.
>
>
>
> ------
>
>
>
> **Q3: Text format around L84–L85**
>
>
>
> **A:** We appreciate this remark and will carefully revise the writing around L84–L85 to improve clarity and readability.
>
>
>
> ------
>
>
>
> **Q4: Missing discussion of recent conditioned egocentric image/video works [1–3]**
>
>
>
> **A:** Thank you for pointing out these relevant works. These methods focus primarily on exo→ego generation and viewpoint transformation, whereas our work targets **ego-centric predictive modeling conditioned on hand trajectories (and robot poses)**, jointly predicting future **actions and frames** for both human and robotic manipulation. We will explicitly highlight these differences in input assumptions, supervision, and target tasks, and cite [1–3] in the related work section in the revised version.
>
>
>
> ------
>
>
>
> **Q5: More time steps as context (beyond one or two frames)**
>
>
>
> **A:** (1) For short prediction, we predict t=2 from t=1, and t=3 from (t=1, t=2). As shown in Table 5, all methods degrade when moving from one-step (rows 1–4) to two-step (rows 5–8) prediction, but the CoSMo + CCA configuration (rows 4 and 8) consistently achieves the best visual metrics and highest Hand IoU.
>
> (2) For longer-horizon prediction, we provide qualitative results in the supplementary material (files 1.gif, 2.gif, and 3.gif) on BridgeData and RLBench using Ego-PM. These clips illustrate that our model can generate longer sequences with coherent hand–object interactions. We will add explicit discussion of these behaviors in the revised version.
>
>
>
>
>
>
>
> ------
>
>
>
> **Q6: Advantage of CCA vs. bi-directional attention**
>
>
>
> **A:** CCA brings two main advantages over standard bi-directional attention in our setting:
>
>
>
> - **Temporal causality:**
>
>     CCA applies a **causal mask** over the key–value tokens ordered by time. This ensures that the action query at step t attends only to tokens from **past or current** time steps, never from the future. This matches the prediction setting and avoids train–test mismatch that can arise when bi-directional attention implicitly uses information from future tokens.
>
> - **Action-centric fusion:**
>
>     In CCA, the **action embeddings serve as the query**, while visual/text tokens are keys and values. This design explicitly asks: *“Which regions and words are most relevant to this particular action?”* and we show in Table 5 (rows 3 vs. 4 and 7 vs. 8) that adding CCA on top of CoSMo and structured action encoding yields consistent gains in FID, PSNR, EgoVLP/EgoVLP+, and Hand IoU.
>
>
>
> By contrast, bi-directional attention treats all tokens symmetrically, which can dilute the action signal and blur temporal ordering. We will add a short paragraph clarifying this conceptual advantage and how it aligns with our quantitative improvements.

---

> > ### Author Response · Authors · 2025-11-20
> >
> > **Q7: Clarifying the ablation on consecutive predictions**
> >
> >
> >
> > **A:** We appreciate the request for clarification. The consecutive-prediction ablation in Table 5 is structured as follows:
> >
> >
> >
> > - **Rows 1–4 (t = 2):** Single-step prediction. The model predicts the next state at t = 2 from the observed state at t = 1.
> > - **Rows 5–8 (t = 3):** Consecutive (two-step) prediction. Here, the model first predicts state t = 2 from t = 1, and then uses **its own prediction at** t = 2 (instead of ground truth) to predict state t = 3.
> >
> >
> >
> > Thus, rows 5–8 directly measure **error accumulation**: all methods degrade from t=2 to t=3, but the variants with CoSMo and CCA (rows 3–4 vs. 7–8) exhibit a **smaller relative drop** and retain the best performance. We will make this explanation explicit in the text and caption, clarifying that rows 5–8 correspond to rollouts where the model’s previous prediction is fed back into the next step.
> >
> >
> >
> > ------
> >
> >
> >
> > **Q8: LWM exocentric bias vs. fine-tuning on egocentric data**
> >
> >
> >
> > **A:** This is a helpful clarification. Our intention was not to imply that LWM remains purely exocentric after fine-tuning, but rather that:
> >
> >
> >
> > - LWM is **pretrained and architecturally optimized** for exocentric (third-person) scenes and vehicle-centric dynamics.
> > - We then fine-tune it on Ego4D to reduce the domain gap (as indicated by the * in Table 1). This improves its performance, but qualitative results (Fig. 3) show that it still struggles with **fine-grained egocentric cues** such as realistic hand shape, and object-relative manipulation.
> >
> >
> >
> >
> >
> > In the revised version, we will clarify the phrasing: instead of attributing the failure solely to “exocentric bias,” we will emphasize that *even after egocentric fine-tuning, LWM retains biases from its exocentric pretraining and design,* which limit its ability to capture fine-grained egocentric hand–object interactions.

---

> > > ### Author Response · Authors · 2025-11-27
> > > **Look forward to your response**
> > >
> > > We sincerely thank the reviewer for the constructive feedback and thoughtful questions. We appreciate the positive assessment of the unified action–visual predictive formulation, the multi-benchmark generality, and the architectural design of our model. Below we summarize the clarifications and revisions made in response to your comments:
> > >
> > >
> > >
> > > - We clarified the mutual benefit between joint action prediction and future-frame generation, and added explicit discussion supported by quantitative and qualitative evidence (e.g., improvements across structured action encoding, CoSMo, and CCA in Table 5).
> > > - We explained how multi-modal fusion operates when CCA is disabled, and will make this mechanism explicit in the revised paper.
> > > - We improved the writing around Lines 84–85 and added missing discussions comparing with recent egocentric generation works [1–3].
> > > - We provided a detailed comparison between CCA and bi-directional attention, highlighting the temporal-causal modeling benefits and the action-centric fusion mechanism.
> > > - We clarified the phrasing regarding the exocentric bias of LWM and explained why limitations persist despite egocentric fine-tuning.
> > >
> > >
> > >
> > > In the revised version, **all modifications will be highlighted in blue** for ease of review.
> > >
> > >
> > >
> > > We hope these clarifications resolve your concerns. If any part remains unclear or if additional analyses or visualizations would be helpful, we would be very happy to provide further details.
> > >
> > >
> > >
> > > We sincerely appreciate your time and valuable comments, and we look forward to your reply.

---

### Official Review · Reviewer_15WE · 2025-10-30

**Soundness:** 3
**Presentation:** 3
**Contribution:** 3
**Rating:** 8
**Confidence:** 3

**Summary:**

This paper addresses a critical gap in egocentric AI: the disjointed modeling of future actions and their visual outcomes. This paper proposes Ego-PM, a novel two-stage framework that unifies action prediction and future video generation conditioned on hand trajectories. The core idea is to use predicted hand motion as an intermediate, guiding representation to ensure that generated future frames are both visually plausible and physically consistent with the intended action. Below are contributions：
1.	First Unified Action-Visual Predictive Model: This is the first model capable of jointly predicting both the upcoming action (as a hand trajectory) and the future visual frames resulting from that action.
2.	Novel Architectural Innovations: Consecutive State Modeling (CoSMo): In Stage I, the model predicts future actions by conditioning on two consecutive previous states. Causal Cross-Attention (CCA): In Stage II, the action embedding from Stage I serves as the Query to perform causal cross-attention over the visual and textual context (Keys/Values).
3.	Demonstrated Generality and Practicality: The same model architecture, without task-specific modifications, achieves state-of-the-art or competitive performance on three distinct benchmarks: Ego4D (human egocentric videos), BridgeData V2 (real-world robot demonstrations), and RLBench (simulated robotic tasks). Crucially, the model requires no external action annotations at inference time, predicting the hand trajectory autonomously, which enhances its practical applicability.

**Strengths:**

Originality: The core originality lies in its novel problem formulation: the joint modeling of egocentric action prediction and future video generation within a single, unified framework. This is a distinct advance over prior work that addressed these tasks separately.
Quality: The technical quality is high. The two-stage architecture is well-motivated and built upon strong, modern components (LLaVA, LDM). The experimental quality is exceptional, featuring rigorous evaluation on three diverse and challenging datasets (Ego4D, BridgeData V2, RLBench).
Clarity: The paper is generally well-structured and clearly written. The logical flow from problem identification to solution and evaluation is easy to follow.
Significance: The work is highly significant for the research community. It directly addresses a critical capability for embodied AI systems: understanding the coupling between an action and its perceptual consequences.

**Weaknesses:**

Analysis of Error Propagation in Long-Horizon Prediction: The experiments primarily focus on predicting one or two frames into the future (e.g., t=2, t=3). A key challenge in predictive modeling is error accumulation over longer sequences. The paper would be significantly strengthened by an analysis of the model's performance degradation when generating longer video horizons (e.g., 20 frames). Does the CoSMo strategy provide robustness against compounding errors compared to baselines?

Computational Efficiency and Latency: The proposed two-stage pipeline, involving a large autoregressive model and an iterative diffusion model, is computationally intensive. For real-world applications like robotic planning, inference speed is critical.

**Questions:**

Long-Horizon Generalization: The model is evaluated on very short-term predictions. Could the authors demonstrate or discuss its potential for longer-horizon prediction (e.g., 20 frames)? What are the main failure modes (e.g., object deformation, trajectory drift) when the prediction horizon extends, and how might the architecture be adapted to address them?

Ablation on Action Representation: The action is represented as hand trajectory coordinates for Ego4D and 7D robot poses for Bridge/RLBench. How critical is the specific form of this action representation? Did the authors experiment with other representations, such as a more abstract latent action space (e.g., akin to AdaWorld), and if so, how did it impact performance?

Causal Cross-Attention Analysis: The CCA module is a key innovation. Could the authors provide more analysis or visualization of the attention maps within the CCA? For example, when the action query attends to the visual keys, which parts of the historical frames (e.g., the object, the hand's current position) does it primarily focus on?

Computational Cost and Latency: The paper would benefit from a discussion (and ideally, metrics) regarding the computational cost and latency of the full pipeline compared to the baselines (especially the faster VLA models and single-stage video predictors).

---

> ### Author Response · Authors · 2025-11-20
>
> **Q1: Analysis of Error Propagation in Long-Horizon Prediction**
>
> **A:** (1) Our main experiments already go beyond single-step prediction and explicitly evaluate *consecutive* prediction: we predict t=2 from t=1, and t=3 from (t=1, t=2). This setup naturally exposes error accumulation. As shown in Table 5, all methods degrade when moving from one-step (rows 1–4) to two-step (rows 5–8) prediction, but the CoSMo + CCA configuration (rows 4 and 8) consistently achieves the best visual metrics and highest Hand IoU. This indicates that CoSMo provides improved robustness to compounding errors compared to the baselines under multi-step rollouts.
>
>
>
> (2) For longer-horizon prediction, we provide qualitative rollouts in the supplementary material (files 1.gif, 2.gif, and 3.gif) on BridgeData and RLBench using Ego-PM. These clips illustrate that our model can generate longer sequences with coherent hand–object interactions. We will add explicit discussion of these behaviors in the revised version.
>
> ----
>
> **Q2: Computational Efficiency and Latency**
>
> **A:** Our architecture is designed to reuse **lightweight adaptations of existing pretrained models**, rather than training large models from scratch:
>
> - Stage I fine-tunes a pretrained LLaVA-style autoregressive model using LoRA on short egocentric clips.
> - Stage II reuses a standard image-based LDM, conditioned on only two historical frames.
>
> This design keeps both memory and compute requirements moderate. In the revised version, we will include a small runtime comparison (parameter counts and per-frame latency) against representative VLA and video-prediction baselines under the same hardware and resolution.
>
> ----
>
> **Q3: Ablation on Action Representation**
>
> **A:** Our action representation is deliberately simple yet explicit and directly usable for control:
>
>
>
> - On Ego4D, we use normalized 2D hand bounding boxes (center, width, height), supervised with L1 + GIoU.
> - On BridgeData and RLBench, we use a 7D relative end-effector pose ($\Delta x, \Delta y, \Delta z, \Delta \text{roll}, \Delta \text{pitch}, \Delta \text{yaw}, \text{gripper}$), supervised with L1/BCE.
>
>
>
>
>
> These choices are summarized in Table 7.
>
>
>
> We ablate *how* the action is injected. In Table 5 (rows 1 vs. 2 and 5 vs. 6), we compare (i) encoding actions as plain text tokens versus (ii) using a dedicated action encoder–decoder. The dedicated encoding yields clear gains in both frame metrics (e.g., FID and PSNR) and Hand IoU, indicating that an explicit structured action representation and specialized encoder are important for performance.
>
>
>
> In addition, Table 8 (Appendix) shows experiments where we use **3D body skeletons** as another form of action input. Under this setting, our model still outperforms the prior work PEVA [1] in a fair comparison, suggesting that our framework is compatible with richer action parameterizations while retaining performance advantages.
>
> ----
>
> **Q4: Causal Cross-Attention (CCA) Analysis**
>
>
>
> **A:** CCA is quantitatively evaluated in our ablations. Comparing rows 3 vs. 4 and 7 vs. 8 in Table 5, adding CCA on top of CoSMo and structured action encoding yields the best overall performance: FID further decreases, PSNR and EgoVLP/EgoVLP+ scores improve, and Hand IoU increases (e.g., from 41.4 to 44.3 at t=2). This indicates that aligning visual and textual features with the action query in a temporally causal manner significantly benefits both action prediction and future-frame synthesis.
>
>
>
> In the revised version, we will further complement these quantitative results with **attention-map visualizations** from the CCA layers, illustrating which regions of the historical frames (e.g., the hand and manipulated objects) the action queries focus on.
>
>
>
> Ref:
>
> [1] *Whole-Body Conditioned Egocentric Video Prediction (PEVA)*.

---

> > ### Author Response · Authors · 2025-11-27
> > **Look forward to your response**
> >
> > We sincerely thank the reviewer for the positive assessment and highly encouraging remarks regarding the originality, quality, clarity, and significance of our work. We also deeply appreciate the constructive questions, each of which we have addressed in detail above.
> >
> >
> >
> > To summarize, we have:
> >
> >
> >
> > - Provided an analysis of multi-step error propagation and discussed longer-horizon behaviors, accompanied by qualitative rollouts in Appendix II (Lines 845–866, Fig. 5).
> > - Clarified the design choices behind the action representations and presented ablations demonstrating their importance, as well as compatibility with alternative representations (e.g., 3D skeletons in Table 8).
> > - Strengthened the analysis of the CCA module.
> >
> >
> >
> >
> >
> > In the revised version, **all modifications are highlighted in blue** to make the changes easy to follow.
> >
> >
> >
> > We hope that these responses adequately address your concerns and further clarify the strengths and contributions of our work. If any points remain unclear, or if additional experiments or analyses would be helpful, we would be very happy to elaborate further.
> >
> >
> >
> > We sincerely appreciate your time and constructive feedback, and we look forward to your reply.

---

### Official Review · Reviewer_ivaC · 2025-10-31

**Soundness:** 2
**Presentation:** 2
**Contribution:** 2
**Rating:** 2
**Confidence:** 3

**Summary:**

is a tow-stage framework for action anticipation and frame generation coupling VLM and LDM through Causal Cross-Attention (CCA). VLM predicts action trajectories that then condition LDM to generate future frames. CCA implements cross-attention with future masking, and is applied to encode and fuse video, text and action for conditioning future frame generation.

**Strengths:**

The design is intuitive and leads to an architecture that achieves sota result when compared to few recent baselines.

**Weaknesses:**

Causal Cross Attention, a main element upon which the contribution of the paper is build, is not particularly novel: standard cross-attention with causal masking.

Why the query in CCA is chosen as the embedding of previous action is not well motivated. It is just adopted this way.

There is no feedback loop that enforces generated frames to contain generated action trajectories following conditioning action trajectories.

Overall, there is no particularly significant novelty. The method is two-stage and not integrated, and there is no grounding loop in the diffusion with the action trajectory conditioning. Loss terms for training are standard or previously published.

It is not clarified in the paper why generating future frame in a 2-stage approach without grounding (see above) should be beneficial. For what should the generated video be used?

Tables are spread around the second part of paper, and the arrangement of frames in figure 3 could be improved. Figure 3 contains only four generated, sparse frames and this makes it difficult to assess the quality of generated future video.

**Questions:**

Please address any of the weaknesses identified above.

---

> ### Author Response · Authors · 2025-11-20
>
> **Q1:** *Causal Cross Attention is not novel; overall novelty is limited. The method is two-stage without integration, and no grounding loop exists between diffusion and action trajectory conditioning. Loss terms are standard.*
>
>
>
> **A:**
> We apologize for the misunderstanding. Our contributions do not rely on Causal Cross Attention (CCA) as a novelty point. Instead, our main innovations are:
>
>
>
> 1. **CCA is not claimed as a contribution.** It is a standard mechanism used only for conditioning future-frame diffusion on historical information.
>
>     Our key contribution is the **Consecutive State Modeling (CosMo)** strategy, which explicitly predicts hand trajectories in a temporally consistent manner.
>
> 2. We propose the **first unified model** that jointly predicts both upcoming actions and future visual frames. Previous works addressed these tasks independently.
>
> 3. Our model is the **first unified predictive framework applicable to both egocentric human videos and robotic manipulation**, showing strong performance on Ego4D, BridgeData, and RLBench.
>
>
>
> These elements, rather than CCA itself, constitute the novelty of our work.
>
>
>
> ------
>
>
>
> **Q2:** *The choice of action embeddings as the query in CCA is not well motivated.*
>
>
>
> **A:**
> Our intention is to use **action embeddings as the query** and visual/language features as key–value pairs so that the model can selectively attend to visual or textual information most relevant to the current action. This leads to stronger multi-modal conditioning for subsequent frame generation.
>
> We will clarify this design motivation in the revised version.
>
>
>
> ------
>
>
>
> **Q3:** *There is no feedback loop enforcing that generated frames contain the generated action trajectories.*
>
>
>
> **A:**
> Thank you for pointing this out. Our current CosMo training strategy (Appendix F) trains the unified model in a step-by-step manner without a feedback loop, primarily to maintain computational efficiency.
>
> Introducing such feedback during training is possible but would increase cost. We will include an additional ablation in the revised paper to examine this option.
>
>
>
> ------
>
>
>
> **Q4:** *It is unclear why a two-stage approach without grounding is beneficial. What is the purpose of generating future video?*
>
>
>
> **A:** Predicting both actions and future frames simultaneously is challenging without additional inputs (e.g., starting 3D body skeletons as required by PEVA [1]). Our two-stage unified design allows:
>
>
>
> - action prediction using only visual–language inputs, and
> - future-frame generation conditioned on predicted actions, without requiring extra information at inference.
>
>
>
>
>
> This yields a practical pipeline for real-world settings: given **only initial frames and language instructions**, the model predictively outputs both **future actions** and **future visual outcomes**. This ability to forecast visual consequences is valuable for planning, anticipation, and safety in embodied agents.
>
>
>
> ------
>
>
>
> **Q5:** *Tables are scattered; the layout of Figure 3 is suboptimal, and only four sparse frames make evaluation difficult.*
>
>
>
> **A:** Thank you for the feedback. We have provided three consecutive generated clips in the supplementary material to better illustrate temporal consistency. We will refine the layout, add more frame samples, and restructure tables for improved clarity in the revised draft.
>
>
>
> ------
>
>
>
> **References**
>
> [1] *Whole-Body Conditioned Egocentric Video Prediction (PEVA)*.

---

> > ### Author Response · Authors · 2025-11-27
> > **Look forward to your response**
> >
> > We sincerely thank the reviewer for the detailed feedback and thoughtful questions. We have carefully addressed each of the identified weaknesses point-by-point in our responses above and revised the manuscript accordingly. To summarize, we have:
> >
> >
> >
> > - Clarified that CCA is not positioned as a novelty, and highlighted our actual contributions—particularly the proposed Consecutive State Modeling (CosMo) strategy and the first unified framework capable of jointly predicting both future actions and frames across both egocentric and robotic settings.
> > - Provided stronger motivation for using action embeddings as the query in CCA, and clarified the design rationale in the revised paper.
> > - Discussed the feasibility, trade-offs, and potential added cost of incorporating a feedback or grounding loop, and outlined our plan to include an additional ablation.
> > - Explained why a two-stage design without external 3D grounding is practical and beneficial for real-world predictive settings where only 2D visual-language context is available.
> > - Improved qualitative visualizations, added more generated video clips in the supplementary material, and revised figure and table layouts to enhance clarity.
> >
> >
> >
> >
> >
> > We hope that these clarifications address your concerns. In the revised version, **all changes have been highlighted in blue** for ease of review.
> >
> >
> >
> > If there are any remaining questions or if you would like us to provide additional analysis, we would be very happy to elaborate further.
> >
> > We sincerely appreciate your time and constructive feedback, and we look forward to your reply.

---

### Official Review · Reviewer_qvyv · 2025-11-01

**Soundness:** 3
**Presentation:** 3
**Contribution:** 3
**Rating:** 4
**Confidence:** 3

**Summary:**

The authors propose a method which, given the action history, visual input and language prompt of either an egocentric video or a robot task rollout video, simultaneously generates imagined action rollouts and, conditioned on these, imagined future video frames. The method makes use of a novel cross-attention scheme to condition a latent diffusion model for the future frame synthesis.

**Strengths:**

The design choices of the method are thoroughly ablated in Table 5. The method shows strong performance against baselines in Tables 2 and 3. The inclusion of both a robotic dataset and a human hand dataset showcases the usefulness and versatility of the method.

**Weaknesses:**

The submission could have benefited from the inclusion of more qualitative examples in the supplementary material, comparing the imagined and real future frames. This is crucial to assess the quality of the generations.

I believe Table 3 should be split into two separate tables, evaluating frame prediction and action prediction separately.

In Table 4, as it stands, the method is strongly outperformed by two baselines depending on 3D point clouds. An interesting comparison would be running the baselines without providing them the additionally required 3D point cloud (e.g. by passing dummy values), to assess the extra gain from its inclusion. I am willing to increase my score if a sensible answer is provided to this point.

In the paper, please clarify how you calculate the hand IoU. It took me a long time to find the detail that you use hand bounding boxes in the supplement. I believe this information is important enough to be included in the main paper.

A bounding box is arguably far less useful than, for instance, a wrist pose or even hand pose, either in 2D or 3D. Thus, I would encourage the authors to study predicting more informative hand representations for egocentric videos.

Table 1 and Table 2 could benefit from the inclusion of more baselines.

**Questions:**

Why is "Ours" bolded for LPIPS in Table 3 when it is outperformed by This&That?

How do you calculate the GT hand bounding boxes when Ego4D does not provide such poses out-of-the-box? Could the baseline Hand IoU in tables 2 and 3 be calculated by using that same method on the imagined future frames of the baselines?

---

> ### Author Response · Authors · 2025-11-20
>
> **Q1: More qualitative examples**
>
> **Response:**
>
> Thank you for the suggestion. We have included three qualitative examples generated on the BridgeData dataset for robotic manipulation tasks in the supplementary material, which already demonstrate the effectiveness of our method. Following your feedback, we will revise the paper to add more qualitative comparisons and richer visualizations.
>
>
>
> ------
>
>
>
>
>
>  **Q2: Table 3 should be split into two tables**
>
>
> **Response:**
>
> We appreciate the comment. Due to page limitations in the initial submission, we combined the two evaluation dimensions (frame prediction and action prediction) into a single table. In the revised version, we will split them into two tables to improve clarity and readability.
>
>
>
> ------
>
>
>
>
>
>  **Q3: Explanation for why two 3D point-cloud baselines outperform our method. Can a fair comparison be run?**
>
>
>
>
>
> **Response:**
>
> **(1) On the performance gap:**
>
> 3D-based robotic manipulation methods (e.g., RVT [1]) reconstruct 3D scenes, render multi-view images, predict heatmaps, and then project these into 3D action coordinates. Although these pipelines achieve high performance, they rely on **costly 3D annotations**, accurate calibration, and **computationally expensive rendering**.
>
> Our method aims to provide a lighter-weight alternative by leveraging **existing 2D annotations** (hand locations for ego-centric videos and arm positions for robot data), avoiding the need for additional 3D supervision or heavy pre-processing.
>
>
>
> **(2) On fairness of comparison:**
>
> 3D-based baselines fundamentally require 3D geometric inputs and cannot operate directly on our 2D annotation modality. Therefore, simply passing a null or placeholder value to these baselines would not produce meaningful results. This makes a strict one-to-one comparison under identical input modalities infeasible.
>
>
>
> ------
>
>
>
>
>
>  **Q4: How is hand IoU calculated? What if Ego4D does not provide hand poses directly?**
>
>
>
>
>
> **Response:**
>
> **(1) IoU computation:**
>
> We treat the hand location as a 2D bounding box. Ego4D provides hand-related annotations from which we extract ground-truth bounding boxes. Our Ego-PM module predicts hand bounding boxes, and we compute the hand IoU using the GIoU metric [2].
>
>
>
> **(2) Ego4D annotations and missing hands:**
>
> During training, we select the subset of Ego4D that contains bounding-box-level hand annotations. About 95% of the selected frame tuples contain at least one visible hand. For frames with no visible hands, we pass a null value to the action encoder.
>
> Crucially, during **inference**, our model directly predicts the trajectory (hand position) without requiring any external annotation, which distinguishes us from traditional VLA methods that depend on ground-truth hand or pose inputs.
>
>
>
> ------
>
>
>
>
>
>  **Q5: A bounding box is less informative than wrist or hand pose. Why choose boxes?**
>
>
>
>
>
> **Response:**
>
> Thank you for raising this point. Indeed, richer annotations such as wrist or full hand pose can provide more detailed geometric cues. Our objective in this paper is to explore a **simple yet effective** form of supervision that generalizes well across datasets and tasks.
>
> Bounding boxes offer a lightweight way to capture hand trajectories while maintaining ease of annotation and training stability. Moreover, our pipeline is **general**: as shown in Appendix E (page 16), it naturally supports more detailed signals such as 3D body skeletons on the Nymeria dataset.
>
> We also find that more fine-grained pose supervision typically requires more training data and may lead to optimization difficulties. In future work, we will provide ablations across different annotation granularities.
>
>
>
> ------
>
>
>
>
>
> **Q6: Table 1 and Table 2 could include more baselines**
>
>
>
>
>
> **Response:**
>
> Thank you for the feedback. We agree that additional baselines would strengthen the comparison and will include more baselines in the revised version.
>
>
>
> ------
>
>
>
>
>
>  **Q7: Why is “Ours” bolded for LPIPS in Table 3 despite being outperformed by This&That?**
>
>
>
>
>
> **Response:**
>
> We apologize for this typographical mistake. We will correct the formatting in the revised manuscript.
>
>
>
> ------
>
>
>
>
>
>  **References**
>
>
> [1] *RVT: Robotic View Transformer for 3D Object Manipulation.*
>
> [2] *Generalized Intersection over Union: A Metric and A Loss for Bounding Box Regression.*

---

> > ### Author Response · Authors · 2025-11-27
> > **Look forward to your response**
> >
> > We sincerely thank the reviewer for the constructive feedback and valuable suggestions. We have carefully addressed each question in detail above and revised the manuscript accordingly by:
> >
> >
> >
> > - Adding additional qualitative examples in Appendix II (Lines 845–866, Fig. 5).
> > - Clarifying key implementation details (Lines 808–809).
> > - Providing a more detailed justification and discussion of limitations regarding comparisons with 3D point-cloud baselines (Lines 308–318 in the revised submission).
> > - Elaborating on the choice of hand bounding boxes versus more fine-grained pose annotations (see results in Table 8).
> > - Correcting minor formatting issues (e.g., bolding in Table 3).
> >
> >
> > In the revised version, all modifications are clearly highlighted in blue for ease of review.
> >
> >
> > We hope these clarifications sufficiently address the reviewer’s concerns.
> >
> > If any points remain unclear, or if further analyses or ablations would be helpful, we would be happy to provide additional details in the discussion or in the final revision.
> >
> >
> >
> > We sincerely appreciate your time and thoughtful comments, and we look forward to your response.

---

### Note · Authors · 2026-01-26

I have read and agree with the venue's withdrawal policy on behalf of myself and my co-authors.

---

### Meta-Review · Area_Chair_CN2S · 2026-01-10

**Summary:**

Reviewers are divided on this paper. **Reviewer 15WE** is positive, but **Reviewers qvyv, ivaC, and LcVE** recommend rejection. This work tackles an interesting problem by jointly modeling action prediction and frame generation conditioned on hand trajectories. While the setup is promising and involves three datasets, the reviewers raise serious concerns about technical novelty and experimental depth.

Several common issues emerged during the discussion. **Reviewer ivaC** argues the Causal Cross-Attention (CCA) is just standard cross-attention with a mask. They see the two-stage structure as a simple pipeline that lacks meaningful integration or feedback. **Reviewer qvyv** notes missing 3D baselines and wants better qualitative comparisons. They also questioned the hand IoU calculation and the use of simple bounding boxes. **Reviewer LcVE** remains unpersuaded that joint prediction actually helps both tasks and asks for better positioning relative to recent generation work.

The rebuttal clarifies that CCA is not the main contribution, pointing instead to Consecutive State Modeling (CoSMo). The authors also explained that hand IoU comes from Ego4D bounding boxes and added ablations to support the joint prediction claims. These responses address some factual points but do not fix the underlying concerns about novelty. The coupling between stages remains weak and the experiments lack the depth needed to prove the model handles long-horizon behavior. Given the score spread of 2, 4, 4, and 8, the paper does not currently meet the ICLR bar.

**Reviewer Concerns:**

**Reviewer qvyv** focuses on the experimental evaluation. They asked for more qualitative examples and a clearer separation of metrics for frame versus action prediction. They also requested stronger baselines and more complete comparisons in Tables 1 and 2. The reviewer is specifically concerned that simple bounding boxes are not as informative as richer hand pose representations.
The authors clarified the focus on CoSMo and explained why 3D point cloud baselines are not a direct comparison for their 2D setup. They also provided more qualitative examples and implementation details for the IoU computation. These answers help, but they do not resolve the broader concern about the strength of the baselines. The reliance on simple hand boxes over richer representations remains a weakness in the experimental design.

**Reviewer ivaC** is primarily concerned with novelty and the integration of the two stages. They argue that CCA is a standard module and that the choice of queries lacks motivation. There is no clear feedback mechanism to ensure generated frames match the predicted action trajectories. This reviewer also notes that the loss functions are standard and the qualitative analysis is sparse.

In the rebuttal, the authors emphasized CoSMo and joint modeling over the CCA module. They added qualitative examples to show how joint training should benefit both tasks. However, the core criticisms regarding limited architectural novelty and the weak coupling between stages remain. The paper still lacks a principled way to enforce consistency between actions and frames.

**Reviewer 15WE** is the most positive reviewer. They like the unified treatment of action prediction and video generation. Their main concern is the short temporal horizon of the experiments. They asked for an analysis of error propagation over 20 frames to see if CoSMo helps mitigate compounding errors.

The authors pointed to Table 5 to show the benefits of joint prediction and argued that CoSMo improves robustness. However, they did not provide the requested long-horizon quantitative analysis. While this reviewer remains supportive of the direction, the questions about long-term stability and error accumulation are not fully answered.

**Reviewer LcVE** questions the evidence for mutual benefits in joint modeling. They find the role of the CCA mechanism unclear and ask why standard bi-directional attention would not work. The reviewer also noted missing discussion of recent egocentric generation literature and asked for clarification on the context window size.

The rebuttal uses Table 5 to argue for joint benefits and explains the design choices for CCA. These clarifications are useful but the overall evidence remains thin. The advantage of CCA over simpler attention is not demonstrated convincingly. The discussion of related work also remains incomplete regarding the latest generative models.

**Reviewer Scores:**

- **Reviewer qvyv (Original: 4 → Predicted: 4)**
  The rebuttal clarifies the IoU computation and adds qualitative examples. These points resolve specific questions but the main concerns about baseline strength and simple hand representations remain. I expect the score to stay at 4.

- **Reviewer ivaC (Original: 2 → Predicted: 2)**
  The authors downplayed CCA to focus on CoSMo, but this does not fix the lack of architectural novelty. The weak coupling between the two stages is a fundamental issue that the rebuttal did not resolve. The score will likely remain a 2.

- **Reviewer 15WE (Original: 8 → Predicted: 8)**
  This reviewer likes the problem setup and the experimental breadth. The lack of long-horizon experiments is a drawback, but it is unlikely to flip their positive stance. I expect the score to remain an 8.

- **Reviewer LcVE (Original: 4 → Predicted: 4)**
  The rebuttal provides some useful clarifications on joint prediction and formatting. However, the evidence for the claimed benefits of joint modeling is still not strong enough. The positioning against recent work is also still a concern, so the score should remain a 4.

---

### Decision · Program_Chairs · 2026-01-26

Reject